# Development of an Adaptive Computer-Aided Soft Sensor Diagnosis System for Assessment of Executive Functions

**DOI:** 10.3390/s22155880

**Published:** 2022-08-06

**Authors:** Katalin Mohai, Csilla Kálózi-Szabó, Zoltán Jakab, Szilárd Dávid Fecht, Márk Domonkos, János Botzheim

**Affiliations:** 1Bárczi Gusztáv Faculty of Special Needs Education, Institute for the Psychology of Special Needs, Eötvös Loránd University, Ecseri út 3, 1097 Budapest, Hungary; 2Department of Artificial Intelligence, Faculty of Informatics, Eötvös Loránd University, Pázmány P. Sétány 1/A, 1117 Budapest, Hungary

**Keywords:** computerized adaptive testing (CAT), soft-sensor based diagnosis, executive functions, neurodevelopmental disorders

## Abstract

The main objective of the present study is to highlight the role of technological (soft sensor) methodologies in the assessment of the neurocognitive dysfunctions specific to neurodevelopmental disorders (for example, autism spectrum disorder (ASD), attention deficit hyperactivity disorder (ADHD), and specific learning disorder). In many cases neurocognitive dysfunctions can be detected in neurodevelopmental disorders, some of them having a well-defined syndrome-specific clinical pattern. A number of evidence-based neuropsychological batteries are available for identifying these domain-specific functions. Atypical patterns of cognitive functions such as executive functions are present in almost all developmental disorders. In this paper, we present a novel adaptation of the Tower of London Test, a widely used neuropsychological test for assessing executive functions (in particular planning and problem-solving). Our version, the Tower of London Adaptive Test, is based on computer adaptive test theory (CAT). Adaptive testing using novel algorithms and parameterized task banks allows the immediate evaluation of the participant’s response which in turn determines the next task’s difficulty level. In this manner, the subsequent item is adjusted to the participant’s estimated capability. The adaptive procedure enhances the original test’s diagnostic power and sensitivity. By measuring the targeted cognitive capacity and its limitations more precisely, it leads to more accurate diagnoses. In some developmental disorders (e.g., ADHD, ASD) it could be very useful in improving the diagnosis, planning the right interventions, and choosing the most suitable assistive digital technological service.

## 1. Introduction

Technological development from the second half of the 20th century on has contributed to the development of the sciences of the mind in at least two ways. First, the advent of computers gave rise to a theoretical breakthrough in understanding cognition as fundamentally information processing. This insight was reinforced, from the very beginning (e.g., [1,2]) by various approaches to computational modeling of cognitive functions. Second, a great number of tools and applications such as brain imaging, data processing, and other computerized and soft-sensor-based forms of data collection came in to support empirical studies and hypothesis testing. One part of this development was that of cognitive neuropsychology, which has, among other things, led to a deeper understanding of neurodevelopmental disorders [3,4]. The rapid progression of infocommunication technology has contributed to the process of fine-tuning psychometric tools based on neuropsychology [5,6]. At present, traditional paper-and-pencil tests are giving way to their computerized counterparts in neuropsychological practice, just like they do in many other areas.

Paper-and-pencil tests are the name given to the various measurement procedures developed to assess psychological functioning, despite the fact that only some of the tests currently in use require the use of paper and pencil. The paper-and-pencil tests are designed to explore objective performances and usually have a fixed formed, linear arrangement, i.e., each assessment follows the same strict procedure ([7]).

## 2. Classical or Technology-Based Testing

Technology-based assessment (TBA) can involve the use of any information and communication technology (ICT) device, such as a laptop or desktop computer, smartphone, tablet, and various applications running on these devices. Thus technology-based testing includes computer-based assessment as well as e-testing (for review see [8]). TBA simplifies the test assessment and administration and results in more valid and reliable measurement data. It is also more time- and cost-efficient, provides immediate assessment and feedback, eliminates data loss, and allows the development of interactive exercises, multimedia elements, and simulations [7,9,10].

The computer implementation of paper-and-pencil tests can take place at several levels, exploiting different potentials. On the one hand, existing paper-and-pencil tests can be converted into digital form without virtually any change [11]. Using the technology of soft sensors at this level offers a number of benefits, such as immediate scoring and feedback, and even higher motivational power. On the other hand, traditional tests can also be implemented in such a way that meanwhile the assessment tool is extended (e.g., by including multimedia tools, by collecting additional information such as reaction time, eye movement tracking, etc.), which allows the simultaneous handling of stimuli of multiple modalities from both the input and the output side. Thirdly, a well-developed task bank can be created, which can be the basis for automatic item generation. This creates an opportunity to give participants different tasks of the same difficulty. In assessing individual variations in cognitive and emotional development, a much more specific and fine-tuned diagnosis can be made at this level [6,12]. At the highest level, a so-called adaptive testing procedure can be implemented by introducing novel test algorithms using fully parameterized task banks. This allows the creation of tasks that match the participants’ abilities so that they receive tasks with the greatest diagnostic power in their case. All of this would be unachievable in linear paper-and-pencil testing. So adaptive testing exceeds the functionality of the original test and significantly increases the diagnostic sensitivity of a given test [10,13,14,15,16]. For example, dimensions of cognitive capacity and their limits can be measured more accurately, allowing more sensitive testing procedures to be developed. This may help to clarify the diagnosis of certain disorders (e.g., Parkinson’s disease or developmental disorders such as ADHD or dyslexia) [5,17,18,19]. This last level is known in the literature as computerized adaptive testing (CAT), and it has the greatest potential [14,20,21]. In our interpretation, CAT and computer-based testing (CBT) are not two dichotomous measurement systems, but there are different levels of CBT. In our opinion, CAT is the highest form of CBT with the most potential and diagnostic power. The essence of CAT is to create an individualized test situation. The software immediately evaluates the person’s answer, which determines the difficulty of the next task, and the participant is given items from the task bank that match their estimated ability level. To achieve this, two key components are needed: (1) a precisely calibrated item bank, and (2) a suitable algorithm that can estimate the participant’s ability level during testing. For the first, that is, calibrating item difficulty, we used item response theory (IRT). For the second component we constructed an algorithm to choose the subsequent item based on the participant’s performance on earlier items (see below for details) [13,22,23]. In addition to its many advantages, CAT has some disadvantages as well. It can aggravate unequal opportunities, as the lack of experience and the unfamiliarity with technology can have a negative impact on results. Using technology-based assessment care must be taken to avoid the disadvantage of certain groups. At the same time, it is worth considering whether the lack of experience can be eliminated by the widespread use of technology [9].

Another disadvantage is the high start-up costs. Developing a calibrated task bank of hundreds of items is time-consuming and costly [24]. In the long run, however, the benefits listed above will offset the significant initial resource inputs, so that significant savings can be made later.

Technology-based testing also raises a number of psychometric questions: although technology-based procedures are more and more widespread, before they replace traditional paper-and-pencil tests, it is necessary to examine whether the two different methods measure the same construct in psychological terms, i.e., whether they produce equivalent results to traditional tests. Therefore, similar results are expected for paper-and-pencil and technology-based tests, unbiased by any medium effect. (The type of the test used in the study—paper-pencil vs. technology-based procedure—should not result in significant variance.) However, this also raises the question of whether it is necessary for the results to be equivalent at all. Future research and impact studies will provide the answers to these questions [8,11].

## 3. Technology and Neurodevelopmental Disorders

Using tests for assessing the characteristics of atypical development is common practice in almost all developed (OECD) countries, but the purpose and method of assessment and the range of people assessed varies considerably [25,26]. There is a general trend for digital test developments to appear primarily in the assessment of a wider student population for screening and measuring more generalized knowledge (e.g., classroom participant tests, knowledge level, and academic performance). In the case of children and learners with special educational needs, digital test development efforts tend to focus on specific, more delimited functional areas (e.g., attentional and executive skills). Technology-based measurement and assessment offer a number of opportunities for the identification of developmental disorders. The technology enables the specialist—educator, teacher, psychologist, doctor—to make a more efficient (e.g., less time-consuming, more valid, more reliable) diagnostic decision, to determine whether a particular disorder can be diagnosed in a certain participant, and makes it possible to assess personality, reveal knowledge or ability profile. For the person concerned, the indirect benefit is that the diagnosis is made more quickly, the results of the diagnosis or assessment are more precise and reliable, so the person can have the most suitable support and education adjusted to his/her personal and ability profile [27].

Overall, the countless possibilities of the digital interface (tablet, smartphone, laptop, etc.) enable an optimal, cost-effective, and more precise investigation of atypical patterns in the cognitive architecture [6]. It is also important to stress that digital assessment of atypical development takes into account well-known standards regarding service and development [26,28]. One of these standards is not to develop separate assessment tools for the typical and for the atypical development, but to create “design for all”, so the main universal concept is to ensure equal access and use [29,30], which of course includes needs-based design.

## 4. The Assessment of Executive Functions (EF) in Neuropsychology

In many cases, neurocognitive dysfunctions can be detected in neurodevelopmental disorders (ASD, ADHD, specific learning disorder), some of them having well-defined syndrome-specific clinical pattern [31,32]. Many evidence-based neuropsychological batteries are available for identifying these domain-specific functions. Some of these atypical cognitive functions, although in different patterns, are present in all developmental disorders, such as the executive functions [33,34,35]. Executive functioning is an umbrella term for the general component of all complex cognitive processes that play a key role in solving novel, difficult and goal-oriented tasks, and in flexibly adapting to environmental changes [36,37]. In fact, the function of executive processes is the differential coordination of mental functioning, the coordination of psychological processes along the lines of perception, emotion, cognition, and execution [37]. It can be defined as a set of capabilities that allows you to represent distant goals, develop plans for achieving the goals, organize and control the cognitive and psychological functioning and behavior, monitor the environment, and, if it is necessary, do flexible modification on the developed plans [38]. This ensures the regulation of behavior, including self-regulation [39] and, as the authors of [38] point out, this is what distinguishes humans from the instinctive and reflexive responses of animals. Although there is no consensus on the precise definition of executive functions, most authors state the multidimensional nature of EF, differed by the number of components, processes, and complexity. Among the components of executive function can be mentioned the inhibitory control or response inhibition, working memory, updating, and shifting [40,41].

## 5. Goal of the Present Study

The aim of this paper is to present CAT-based procedures of a well-known and widely used neuropsychological test assessing executive functions as a soft sensor.

Although most neuropsychological tests of executive functions are already computer based, our search did not reveal any studies presenting CAT-based versions of these tools. To understand why, we need to know that the development of psychometric tests used in the assessment of neurodevelopmental disorders is based mostly on classical test theory. Classical test theory (CTT) focuses on total test score; it characterizes item difficulty as the relative frequency of examinees who answer an item correctly and derives item discriminability from item-total correlations. Item response theory, on the other hand, focuses on individual items characterizing item difficulty and examinee ability on the same scale whose units are logits (log odds of the probability of correct response). An important advantage of IRT over CTT is that the former assures that the joint difficulty–ability scale is at interval level [13].

In general, most neuropsychological tests are based on classical test theory. At the same time, the development of item banks at considerable financial cost seems unnecessary, as the main function of the tests is to help the diagnosis and differential diagnosis, which can be achieved best by increasing the sensitivity of the items rather than by creating a large item bank.

Based on adaptive test theory, we developed the Tower of London Adaptive Test and the Corsi Adaptive Test, measuring different aspects of executive functions, the former is used for assessing planning and problem-solving ability, while the latter is used for assessing visual working memory. In this paper, we present the Tower of London Adaptive Test (ToLA). Our aim was to assess the level of ability as sensitively as possible and do so in a relatively short time. This helps us, for example, eliminate the effect of attention disorder (i.e., by abolishing the need to maintain attention for an extended time) from planning and strategy-making ability, thereby enabling a more accurate differential diagnosis. In order to achieve this, the focus was not only on creating an item bank but on the analysis of the characteristics (e.g., solution time, number of steps required to solve a given item) of initially well-thought-out items and the use of these results.

In addition to the above-mentioned benefits, the synergy of this kind of adaptive solution with the soft sensor system also has a higher added value, hence a very precise measurement is enabled this way in terms of the interactions which is often time and effort-demanding from the specialist’s side during or after the tests. Additionally, the evaluation phase is made less burdensome for the specialist by only presenting the (often calculated) measures necessary during a diagnosis. In this respect, this study has a significant added value and innovative power in the assessment of neurodevelopmental disorders with soft sensors.

## 6. Administering the Tower of London Adaptive Test

### 6.1. Background

The Tower tasks constitute a widely used method for examining planning and strategic thinking as subcomponents of executive functions. Based on the earlier Hanoi Tower task, the London Tower task was developed by [42], for studying the planning processes of patients with frontal lesions. Attaining the goal state using the smallest number of moves often involves a bypass strategy whereby initially the distance from the goal state needs to increase in order for the key moves to become feasible (according to some metrics at least, e.g., how many of the discs are currently located in the stack where they should be in the end state).

In the Tower of Hanoi task, discs of different sizes piled up on a peg from largest at the bottom to smallest on top have to be transferred to one of two similar pegs in such a way that only one disc is moved at a time, and bigger discs must never be placed on smaller ones. The minimum number of moves needed is $2^{n}-1$ for *n* discs in the initial pile.

In the Tower of London task, there are four discs that are the same size but all are differently colored; they are located in stacks of different heights—typically one stack can hold three discs, whereas two others can hold two discs each. The discs are arranged in a starting configuration, and the required goal configuration is also shown; the task is to reproduce the goal configuration from the starting one while never placing more discs in a stack than what it can hold.

The development of the tests in their current form was preceded by a pilot study; the results of which can be found in [43].

The current study is based on the results of an earlier pilot study. In 2015 we had an opportunity to develop computer-based tests for education centers involved in special-needs diagnoses. Since up to that point digital testing had not been applied in Hungary in any systematic way, in developing tests and their framework we had to build on the limited available means at the time. (The pilot study was supported by EU funds, in particular the Social Renewal Operation Programme—project number TÁMOP-3.4.2.B-12-2012-0001.) In this pilot project, we developed reading and language tests along with a few digitalized procedures to measure neurocognitive functions. The software was developed for touch screen tablets and optimized for the Windows 8.1 operation system. The tests were validated on a large sample of atypically developing children in 2014–2015 (please see the Results for detail). The American Psychological Association’s (APA) ethical standards were followed to conduct this study. Written informed consent was obtained from the parents of all participating children. The participants were informed about their anonymity protection as well as about the possibility to withdraw from the study at any point during the process. Every participant was given a code for research purposes. The most important result of this pilot study was a clinical trial, and the development of corresponding test norms (see [43] and Appendix of [43]). Within a few years, however, the instruments and operation system for which these tests were developed became outdated, therefore their regular use grew more and more difficult, even though the need for their application persisted. In the present study, we substantially reworked our earlier computerized tests of executive functions including the Tower of London (TOL) Test. This new version has not been subjected to clinical trial yet, however, here we offer a reanalysis of the TOL data collected by the first version and published earlier. Our purpose in the present study is to create a web-based soft sensor for multiple users which suits the needs of both the users and clinical experts and can be developed further whenever the need arises.

### 6.2. The Framework

As a first step toward transforming classical neuropsychological tests into an adaptive form, we developed a framework that can be applied to several instruments. The framework comprises three major phases. The first is Familiarization, which allows the participant to become familiar with the interface and understand the task, followed by the Assessment level, which aims to assess the capabilities of the participant and is the starting point for the Testing level. Finally, during the Testing phase, the computer generates tasks of different difficulty levels based on the score obtained at the Assessment level (this is the adaptive feature of our algorithm). An overview of the described system can be seen in Figure 1.

This framework provides an opportunity for the participants to familiarize themselves with the test situation before the actual testing, and to train themselves in the way of responding. All of these stages are supported by multimedia elements (including sound files and animation). In the so-called verification phase belonging to the Familiarization phase, the participant receives corrective feedback on the solution of the task, while the examiners can verify the correct understanding of the task. The standardized structure of the test helps the participant in adapting to the tasks. The evaluation is also automated; the different test parameters and scores are displayed in an output file.

### 6.3. Software

The developed software application was optimized for web-based usage for Chromium-based browsers. This ensures great flexibility and the test is not restricted to running on personal computers, it also has the ability to be used with smartphones or tablets, merely an internet connection and a browser are needed.

During the development, an important factor was that the system needs to be designed for usage by professionals as they can properly interpret the test results. Therefore, a Laravel 8 framework-based portal was created around the test that handles the access to the test and its results. When choosing the framework of the portal, it was considered that it should be a stable technology that is widely used this way during further development it is not tied to a specific developer in the future, and due to its widespread use, fewer bugs or vulnerabilities can be expected.

Registered professionals can enroll participants for testing. Participants must be of sufficient age to be allowed by the framework to complete the test. The data of the professionals and the participants can both be modified anytime. The participants are in 1-N relation to the tests that can be completed. The usage of this relation was necessary in order to enable the tracking of the participant’s development. Sensitive data are stored through a cryptographic hash function, which prevents them from being extracted from the database in an event of unauthorized access.

The test itself is a JavaScript-based game with a logger of events. Its framework is based on Phaser 3, which greatly facilitates easy implementation. The essentially open source HTML5 framework, as a built-in functionality, handles various web browsers, while providing sufficient flexibility and not limiting the development. In Phaser, the test’s levels of difficulty are built upon so-called “Scenes” in Phaser. These scenes are accessed by the test via the system’s automation according to the predefined conditions (i.e., whether the participant gave a good or bad answer). At the end of a test, the data is sent to the server at the touch of a specific button by the specialist.

According to the current implementation, the test consists of the following three phases as mentioned in Section 6.2:During the Familiarization phase, an animated tutorial is performed, which is designed to get high attention from the participants’ age group (see Figure 2). At the end of the Familiarization, there are two test cases with feedback. There is no measurement during the Familiarization phase test cases.The Assessment phase has a maximum of four test cases. The four assessment phase tests have different levels of difficulty. Here the predefined stopping condition is a bad answer, the assessment phase ends and the Testing phase begins.Finally, according to the participant’s results, the Testing phase is started at a level defined by the results from the Assessment phase. The number of all test cases is 10×3, there are 3 test cases per level.

During the game, it is considered a wrong solution if the participant does not reach the specified layout or the allowed time expires or the participant runs out of the allowed number of moves. In this case, the program sets an error flag but lets the participant keep on working, does not advance the tests, and does not indicate the failure despite that the test’s answer is already recorded as an incorrect one.

If the participant has solved all the assessment tests then he/she starts the Testing phase from level 7. Otherwise, he/she has to start the level prior to the unsolved level. There must be at least two correctly completed test cases from the three for the participant to pass a level.

Only discs under the inscription “GameSpace” and discs on the top of the stack can be moved. They are moved by dragging them and dropping them to the desired place. The game saves the number of the given level and the flags of the errors. From the start of the test case, all the events are logged in milliseconds (i.e., when a disc is grabbed or is dropped, and when the participant pressed the “Done” button). The program then determines, based on the disc coordinates, whether the layout of the discs is the same as the target layout. The system checks whether there were any error flags. After the evaluation, the system chooses the new level of difficulty according to the algorithm and the participant automatically enters the next test case or the testing is stopped.

The saved data is stored on a server categorized by participant ID. By clicking on the participant’s data sheet, the system returns the results of the completed tests ordered by date and in tabular form. In this case, it is checked whether the requester has the right to access the given data or not: so the participant belongs to the given user, otherwise, the request is refused and data is not sent from the server to the client.

### 6.4. Test Administration

The development of the Tower of London Adaptive Test required the determination of difficulty levels, the assignment of the optimal duration, and the number of steps required to solve the problems, as well as the development of the algorithm. Difficulty levels were determined by the number of steps needed to solve certain tasks. For selecting the most appropriate items for each difficulty level we used computer simulation. Based on simulation it became clear that upwards of the eight-step task, the three-disc space is no longer sufficient. Therefore, from level 1 onwards, we introduced not only the three-disc but also the four-disc task, so that children could get familiarized with both arrangements (see Figure 3).

During the task, the participant had to arrange the discs according to the configuration shown on the top of the screen. Only one disc was allowed to be moved at a time, and the number of discs piled up in a stack could not exceed stack height (see Figure 3). There were ten difficulty levels in total (corresponding to the number of moves necessary to reach the goal configuration) with three different tasks at each level. If the participant successfully solved two out of the three tasks, he/she passed the level and proceeded to the next difficulty level. If the level was not passed, a task from a lower difficulty level followed. Figure 4 summarizes the adaptive testing algorithm that was used for data collection in our earlier study [43].

The duration and number of steps for solving a given problem were limited, and both of them could be monitored on the screen. The test was interrupted if two consecutive levels were not passed.

However, in the original study, we set high limits for both time and number of steps. In particular, at each difficulty level the number of allowed moves was twice that of minimally necessary moves (e.g., 20 for problems that can be solved in ten moves). Evaluating the collected data using these high limits resulted in a ceiling effect (that is, high success probabilities for even the most difficult items). Therefore, in the current reanalysis we set lower maxima on the moves allowed, at each difficulty level. That is, some responses that were eventually successful, but used many more moves than necessary were now not accepted as correct. Put another way, if the required strategic planning was diluted by too much trial and error on the part of the participant, we did not score the performance. Figure 4 shows the new parameter values (see the upper values in the small red rectangles attached to ovals representing difficulty levels).

### 6.5. Analysis and Results

As mentioned above, the development of our present test was preceded by a pilot study [43]. In this study, the data were collected by our adaptive algorithm, and the results were analyzed by classical test theory. In what follows we reassess these data, and analyze them using item response theory.

The original sample consisted of 302 children of whom 214 were typically developing, and 84 had a learning disorder (dyslexia and/or attention deficit). All children were native Hungarians from different settlement types in Hungary. Although the sample was not representative (i.e., not compiled with the assistance of the Central Statistical Office), care was taken for the sample to approximate the characteristics of the population of same-aged children in Hungary, as well as possible. Type of residence, parents’ level of education, and gender ratios were taken into account to this end. The mean age was 9.1 years (sd: 1.0 yr); the youngest participant was 7 years old, whereas the oldest one was 12 years. There were 102 second graders, 91 third graders, and 105 fourth graders in the sample; two participants were in grade 5, and two were in grade 6. Exclusion criteria for the neurotypical sample were premature birth; sensory impairment; any extant neurological syndrome (e.g., epilepsy); taking any medicine affecting the nervous system; non-promotion (repeating a school year); diagnosed need for special education (dyslexia, dyscalculia, ADHD, intellectual disability). Inclusion criteria for the clinical sample were a diagnosed disorder of reading, writing/orthography, or arithmetic ability; a diagnosis of ADHD; taking corresponding medications was not a reason for exclusion. Diagnosis took place according to the Hungarian psychological and special education protocol: assessment was conducted in psychodiagnostic centers based on teamwork.

Data used in this study was collected from 214 typically developing children (age range: from 7.08 years to 11.92 years; mean: 9.00 years, sd: 1 year; 99 females, 115 males).

Each participant completed an adaptive testing procedure involving 30 TOL problems as items. The items were divided into ten broader levels of difficulty according to the minimal number of moves necessary to reach the solution (1 to 10; three items at each level). The probability of a correct response was readjusted by lowering limits on the maximum number of moves at each level; the new limits are summarized in Table 1.

Theoretically, there is no guarantee that two problems with the same number of minimally necessary moves are equally difficult from a psychological point of view. Therefore there is no reason to expect batches of three items to have identical (or nearly identical) difficulty indices (theta values in item response theory used below), and for this reason, initially, each item was entered in the analysis. However, five of the items were so easy as to produce extreme, and in one case, obviously incorrect difficulty estimates. (The item with a grossly incorrect theta estimate was one with 100% correct responses. This item was subsequently omitted from the analysis along with four others which resulted in 212 or 213 correct responses out of 214).

Note also that the guessing parameter can be reasonably ignored in our case as there is a vanishingly small chance for a participant to come up with the correct solution to a TOL problem by randomly moving discs. (In fact, for the simplest problems that require only one move, the probability of a random correct response may be up to 1/3, but as the number of necessary moves increases, the probability of correct guessing rapidly falls off. Thus there is no stable level for the probability of correct guessing. Moreover, due to the adaptive procedure used, most correct responses to the simplest problems were granted to participants without actually administering the items to them, so correct guessing did not seem to have played an important role in these cases. In further studies, however, an option is to omit extremely simple problems altogether from the analysis.) Therefore a 2PL, dichotomous items model was suitable for our purposes. We used the Latent Trait Models (ltm) package in R to conduct the analysis [23,44]. (Latent trait models assume that all test items measure the same underlying ability relying on largely the same mental processes. Given that the TOL test comprises tasks of one type, this assumption is reasonable in the present case. Each such model uses at least one parameter, namely item difficulty, which is calculated from the probability of correct response for an item using the sigmoid—or logistic—function. 1PL—that is, one-parameter logistic—models are built on the difficulty parameter alone, 2PL models introduce, in addition, an item discriminability parameter, whereas 3PL models include a third parameter which is the probability of correct guessing [45]). Table 2 shows the model summary.

Item difficulty ranges from −4.89 (Item_1_2) to 0.8017 (Item_10_3), which is not symmetrical around 0, therefore further, more difficult items should be introduced later. Item discrimination is generally good, as 21 out of 25 items have discrimination parameters above or very close to 1; four items are at the margin of acceptability (Item_3_3; Item_5_2; Item_5_3; Item_7_1), and none are very poor. Figure 5 shows the test information function which again indicates that the introduction of further, more difficult items are needed—as the curve is asymmetrical; the present set of items gives us somewhat less information about participants with high ability, although it is good in the range from moderately low through average to moderately high range.

Item fit is also acceptable; 9 out of 25 items had significant Chi-square values (Table 3).

Next, we compared this 2PL model with the 3PL alternative including a guessing parameter expressing the probability that a low-ability participant solves the item correctly. The highest value of the guessing parameter was 0.0024 (Item_3_2), and the lowest one was $10^{-301}$ (Item_10_3), that is, introducing this parameter indeed made no difference. This is also confirmed by the summary of the ANOVA comparing the two models (Table 4). In Table 4, three indicators are reported to compare the two models in terms of how well they fit to the data. Aikake’s information criterion (AIC), and the Bayesian information criterion (BIC) can generally be interpreted only via using pairs of their values; of two models the one with better fit is the one with a smaller AIC or BIC. As can be seen in Table 4, the 3PL model actually exhibits a poorer fit than the 2PL one, however, the difference between them is not significant. The log-likelihood statistic is more stable in this case, corresponding to the absence of a significant difference between these two models.

Although the variability of the item discrimination values (Table 2) indicates that a one-parameter Rasch (or 1PL) model would introduce an undesirable constraint, the comparison between the Rasch model and the 2PL model was also made (Table 5). As the results show these two models do differ substantially, to the advantage of the 2PL model. AIC, BIC, and log-likelihood are all higher for the one-parameter model, indicating poorer fit, and the difference in the goodness of fit between the two models is highly significant.

## 7. Summary

Present technological development can be harnessed to assess the individual needs of children in different areas, including developmental diagnosis and support [46]. To this end, we need up-to-date instruments based on soft sensors to measure individual differences and pair them with suitable interventions when necessary. In the diagnosis of persons with disability, the involvement of technological innovations has become a subject of systematic study. Digital testing is a key component of this process whereby individualized training regimes, as well as population-level monitoring of performance, can be implemented. The soft sensor we developed serves the purpose of educational diagnosis, that is, it helps pedagogues, special education experts, psychologists, and doctors to establish faster and more efficiently whether a certain developmental status obtains for an individual being examined. As part of this, our system measures certain aspects of personality, ability profile, and knowledge. For the examined individual the benefit is that they can have a diagnosis faster, and the results are more reliable than with traditional testing procedures. As a result, the ensuing support and education will be better tailored to personal needs [27]. We need to emphasize, however, that involving technology in diagnosis, and tailoring it to special needs support is a preliminary stage to application in learning environments [6,47]. In maximizing the benefit of using technology we need also to pay attention to minimizing possible risks. Limitations—and future perspectives—of our study are the following. First, with the present version of our test software, we have not collected data yet—this is a task for a future study. Instead, we reanalyzed data collected by an earlier version of our TOL test program. Our results showed that TOL data are amenable to IRT analysis, and the results obtained are satisfactory, indeed, promising. In this respect, our expectations are fulfilled. A second limitation is that so far we have used a fixed adaptive algorithm that did not include a machine learning component. Another objective for further research is to include machine learning in our adaptive algorithm, with the aim of measuring participants’ potential for relatively short-term improvement, not just performance at a given point in time. What a participant can achieve with modest support, as opposed to completely autonomously, is called the zone of proximal development [48]. Using machine learning, it becomes possible to conduct dynamic measurement which, unlike static measurement, supplies information about changes in individual learning strategies, or individual learning potential [49,50]. This information can in turn be utilized in adjusting education to individual needs. This information can be fed back into improving educational effectiveness adapted to the needs. A third limitation, or plan, is to extend our item bank, especially by constructing more difficult items (with more than 10 moves necessary to succeed) as this would tap better into the high ability range of planning and executive functions. Taking the present line of research further, we plan to conduct a systematic comparison of non-adaptive paper-and-pencil tests with their CAT versions to contribute to such benefit-and-risk analyses. We think that info-communication technology-based procedures need to be supplemented by empirical studies of application and effectiveness to become reliable and trustworthy methodologies.

## Figures and Tables

**Figure 1 sensors-22-05880-f001:**
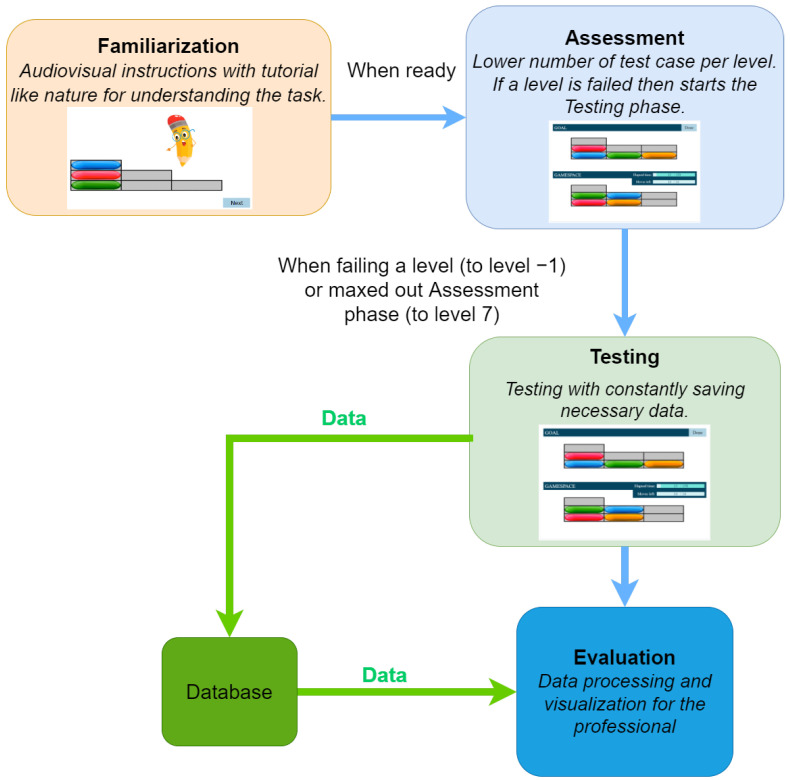
Overview of the system with the three phases described.

**Figure 2 sensors-22-05880-f002:**
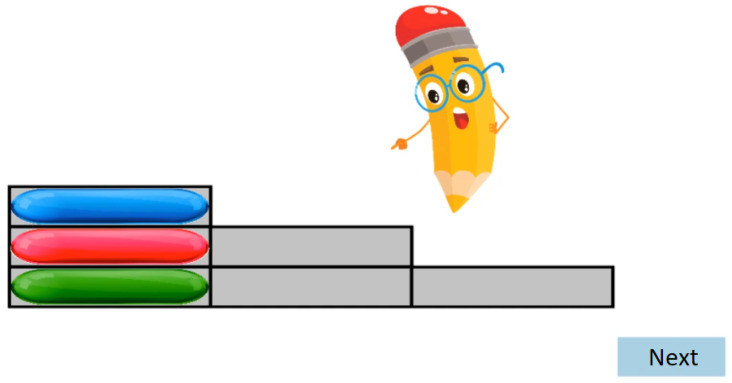
Tutorial with animation during the Familiarization phase.

**Figure 3 sensors-22-05880-f003:**
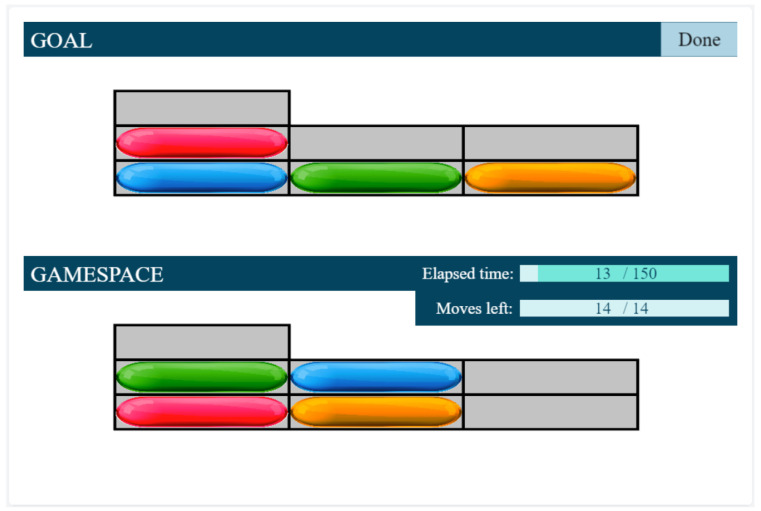
Screenshot of the gamespace during the Tower of London Test.

**Figure 4 sensors-22-05880-f004:**
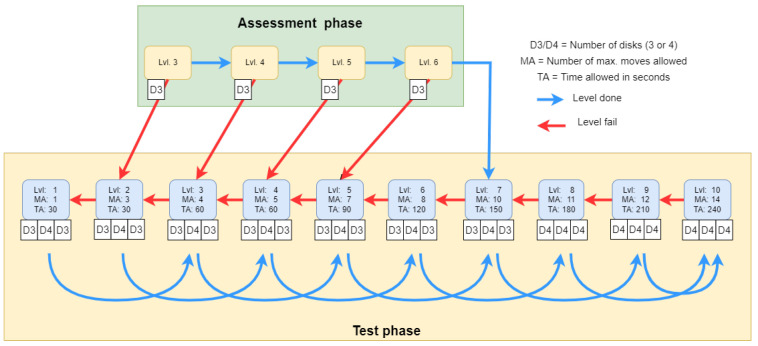
The algorithm of Tower of London Adaptive Test. Blue arrows represent the step if a test level is done, and red arrows the failing of a test. D3 and D4 represent the number of discs in the test case (three discs and four discs respectively). MA denotes the allowed maximum moves, and TA stands for the allowed maximum time.

**Figure 5 sensors-22-05880-f005:**
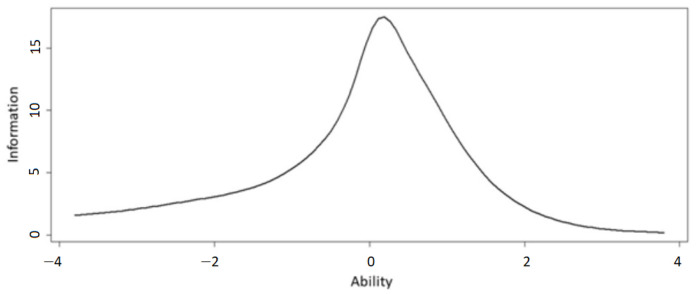
Test information function for the 25 Tower of London items.

**Table 1 sensors-22-05880-t001:** Difficulty adjustment by setting maxima on the number of moves allowed in the Tower of London Test. Exceeding the number of allowed moves for an item resulted in a score of 0 for that item regardless of whether the participant eventually reached the required configuration or not. As a result of this adjustment, the probability of correct response to the easiest item was 0.991, whereas that to the most difficult one was 0.271.

Level (number of minimally necessary moves)	1	2	3	4	5	6	7	8	9	10
Moves allowed	1	3	4	5	7	8	10	11	12	14

**Table 2 sensors-22-05880-t002:** Summary of the 2PL model parameters.

	Difficulty			Discrimination		
	Value	std.err	z.vals	Value	std.err	z.vals
Item_1_2	−4.8904	1.9344	−2.5281	0.9907	0.4987	1.9866
Item_3_1	−4.9173	2.006	−2.4513	1.1036	0.6024	1.8318
Item_3_2	−4.3959	1.4469	−3.0382	1.0487	0.4528	2.316
Item_3_3	−2.957	0.8585	−3.4443	0.6356	0.1981	3.2088
Item_4_1	−2.5104	0.5025	−4.9956	1.0311	0.2511	4.1064
Item_4_2	−2.0959	0.3384	−6.1936	1.3667	0.294	4.649
Item_4_3	−2.1625	0.3542	−6.1046	1.3482	0.2946	4.5768
Item_5_1	−1.4868	0.2948	−5.0429	1.0414	0.2134	4.8809
Item_5_2	−1.1268	0.347	−3.2473	0.6743	0.1731	3.8946
Item_5_3	−0.7598	0.2605	−2.9172	0.7722	0.1812	4.2625
Item_6_1	−0.6452	0.1684	−3.8325	1.3412	0.2423	5.5359
Item_6_2	−0.8772	0.1888	−4.6455	1.3214	0.2396	5.5148
Item_6_3	−0.504	0.1139	−4.4251	2.7899	0.491	5.6826
Item_7_1	0.1914	0.2075	0.9228	0.7322	1855	3.9468
Item_7_2	−0.7521	0.1988	−3.7842	1.0973	0.2121	5.1743
Item_7_3	−0.1788	0.1442	−1.2404	1.2702	0.237	5.3586
Item_8_1	0.2875	0.126	2.2826	1.4433	0.2802	5.1509
Item_8_2	0.2639	0.0895	2.9476	2.9337	0.5436	5.397
Item_8_3	0.1208	0.0788	1.5325	5.3265	1.434	3.7145
Item_9_1	0.5437	0.1192	4.5604	1.7809	0.3563	4.9981
Item_9_2	0.9257	0.1589	5.824	1.7362	0.3887	4.4662
Item_9_3	0.7767	0.1279	6.0739	1.9745	0.4043	4.8842
Item_10_1	0.7677	0.1262	6.0855	1.9243	0.397	4.8475
Item_10_2	0.6361	0.0872	7.2932	3.4896	0.8635	4.0411
Item_10_3	0.8017	0.12	6.6781	2.1448	0.4535	4.7295

**Table 3 sensors-22-05880-t003:** Item fit for the 25 Tower of London tasks. Significant *p*-values are marked with bold.

	$\chi^{2}$	sig.
Item_1_2	8.4019	0.3952
Item_3_1	14.8232	0.0627
Item_3_2	7.9789	0.4355
Item_3_3	13.322	0.1012
Item_4_1	4.9713	0.7606
Item_4_2	3.9275	0.8636
Item_4_3	7.2758	0.5072
Item_5_1	9.1146	0.3327
Item_5_2	23.0207	**0.0033**
Item_5_3	29.703	**0.0002**
Item_6_1	18.1583	**0.0201**
Item_6_2	17.2215	**0.0279**
Item_6_3	10.4865	0.2325
Item_7_1	26.7456	**0.0008**
Item_7_2	19.8636	**0.0109**
Item_7_3	24.3304	**0.002**
Item_8_1	16.5941	**0.0346**
Item_8_2	3.5923	0.8919
Item_8_3	7.1894	0.5163
Item_9_1	9.9894	0.2658
Item_9_2	17.9461	**0.0216**
Item_9_3	8.5134	0.385
Item_10_1	11.2503	0.1879
Item_10_2	7.5525	0.4784
Item_10_3	9.3497	0.3137

**Table 4 sensors-22-05880-t004:** Model comparison: 2PL and 3PL.

Model	AIC	BIC	log.Lik	LRT	df	sig.
2PL	4766.76	4935.06	−2333.38			
3PL	4816.76	5069.21	−2333.38	0	25	1

**Table 5 sensors-22-05880-t005:** Model comparison: Rasch and 2PL.

Model	AIC	BIC	log.Lik	LRT	df	*p* Value
Rasch	4890.91	4975.06	−2420.46			
2PL	4766.76	4935.06	−2333.38	174.15	25	<0.001

## Data Availability

Enquiries concerning data generated or analyzed during this study can be directed to the corresponding author.

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
