# Peer review of "Development of an Adaptive Computer-Aided Soft Sensor Diagnosis System for Assessment of Executive Functions"

_sensors, 2022, doi:10.3390/s22155880_

Round 1

Reviewer 1 Report

The work entitled Development of an Adaptive Computer-Aided Soft Sensor Diagnosis System for Assessment of Executive Functions contains new scientific knowledge and covers a relevant topic. The paper is well written. However, I have some comments that should be addressed before the manuscript could be considered for publication.

Introduction:
- Authors should consider justifying the following affirmation with some citations:

One part of this development was that of cognitive neuropsychology, which has, among other things, led to a deeper understanding of neurodevelopmental disorders. The rapid progression of infocommunication technology has contributed to the process of fine tuning psychometric tools based on neuropsychology.”

I do not know if the following affirmation is correcto r a mistake. If so, authors should further explain it:

“An important advantage of IRT over CBT is that the former assures that the joint difficulty-ability scale is at interval level”

Analysis and results:
From my point of view, the information about the age should be presented in years and month ( to make it more accessible). Also, authors should refer to participants instead of subjects.

Also, how was the selection of the participants? What was the nationality, the educational level of the nationality of the participants among others relevant variables? Did authors had to eliminate participants? Did authors check for previous history of mental problems or learning difficulties?

Summary;
I think that authors should consider further elaborating the discussion/conclusion section. Authors need to further explain the extent to which the objectives and hypothesis were satisfied. How they results relate to previous research. Also, implications of the results found for research and clinical/educational practice could be mentioned. In addition, limitations of the study should be mentioned.

Authors mention that they excluded participants attending to different criteria. In my opinion, it would be interesting to know how many did the exclude for each of the mentioned reasons.

Also, did authors check for previous history of mental health problems and for previous use of healthcare?

Results:

Authors may want to consider including values of statistical analysis (at least showing the p values) in the Table. Maybe changing or generating a new table could be useful to this end.

In addition, authors mention thaty stressed alliance (n = 14, 31%) 153 and disordered alliance (n = 13, 29%)”. However Stressed Alliance and Disordered Alliance do not appear in the table.

From my opinion, it could be worthy to include the differences found between the three types of families in the different measures.

Discussion:

The discussion, from my point of view, could benefit for the introduction of new recent literature (as mentioned in the previous point from introduction).

Also, some lines could be devote to the prevention strategies derived from this study.

Reviewer 2 Report

The authors present an improved version of the application of the Tower of London Test, a neuropsychological test widely used to assess executive functions. In particular, they propose the use of computerized adaptive testing (CAT) theory to improve test application, since the purpose of the CAT is to adapt to the ability level of the examinee and determine the level of difficulty of the next task.

The authors conclude that the results of their proposal improve the diagnostic power and sensitivity of the original test. In addition, they state that the procedures based on information and communication technologies must be complemented with empirical studies of application and efficacy to become reliable and trustworthy methodologies. The work is of excellent quality, but it can improve substantially if they address the following recommendations and questions:

The introduction must be improved, exposing the problem addressed, explaining the objective of the work and the contributions of their work to the field of study.

Section 2 and 3 are very interesting but due to their content and length they should be merged with the Introduction section.

In section 5, the added value and innovative power in the assessment of neurodevelopmental disorders with soft sensors should be highlighted more clearly.

In Section 6.2, they should integrate a schematic image of the framework, where the three phases are detailed.

In Section 6.4, they should explain more about figure 3.

In Section 6.5, it is not explained why Latent Trait Models were used for the analysis.

Reviewer 3 Report

This article is interesting and describes a CAT for a particular context well. Having said that, the article may improve by providing more details on some analysis and methodological aspects, further to the background.

First, there may be an additional reflection on the difference between CAT and CBT. Lines 143-157 provide some details, but for a reader who may not be in the domain, the difference may not be very transparent. Perhaps the authors can consider including a table to outline the difference and how the existing study differs from the previous ones. Simultaneously, since all readers may not be familiar, may clarify what executive functions imply across the paper. 

Similarly, the authors may consider clarifying the steps outlined in 6.2 a bit better. Instead of just texts, a figure or a block diagram will improve the paper's readability. 

Concerning analysis, it's questionable why age is expressed in terms of months rather than years. Also, what sampling procedure was adopted to recruit the 214 participants? Was it random sampling or quota sampling or purposive sampling or similar? The authors may consider clarifying further details.

The adjustment of probability for correct responses, mentioned in Lines 303-5, may be explained further - perhaps showing a sample or an equation may make it clearer.

Similarly, while there are descriptions of the software's functionalities, the technical aspects of how the designed software is a CAT and not a CBT may be better outlined. Does the software use any machine learning technology that makes it adaptive?

 Concerning the presentation of results, technical terms like 2PL and 3PL may be explained - not all readers will be familiar with these terms, and the results may not be fully realised. Tables 4-5 may be better explained. 

Overall, the article may be improved in its presentation.  

Round 2

Reviewer 1 Report

Authors have addressed all my comments. I have no further suggestions.